# Synthesis of Silver Nanoparticles Using Extracts from Different Parts of the *Paullinia cupana* Kunth Plant: Characterization and In Vitro Antimicrobial Activity

**DOI:** 10.3390/ph17070869

**Published:** 2024-07-02

**Authors:** Alan Kelbis Oliveira Lima, Lucas Marcelino dos Santos Souza, Guilherme Fonseca Reis, Alberto Gomes Tavares Junior, Victor Hugo Sousa Araújo, Lucas Carvalho dos Santos, Vitória Regina Pereira da Silva, Marlus Chorilli, Hugo de Campos Braga, Dayane Batista Tada, José Antônio de Aquino Ribeiro, Clenilson Martins Rodrigues, Gerson Nakazato, Luís Alexandre Muehlmann, Mônica Pereira Garcia

**Affiliations:** 1Nanobiotechnology Laboratory, Institute of Biological Sciences, University of Brasilia (UnB), Brasilia 70910-900, DF, Brazil; kelbislima@gmail.com (A.K.O.L.); mgarcia@unb.br (M.P.G.); 2Brazilian Agricultural Research Corporation (EMBRAPA), Embrapa Agroenergy, Brasilia 70770-901, DF, Brazil; jose.ribeiro@embrapa.br (J.A.d.A.R.); clenilson.rodrigues@embrapa.br (C.M.R.); 3Basic and Applied Bacteriology Laboratory, State University of Londrina (UEL), Londrina 86057-970, PR, Brazil; lucmst96@gmail.com (L.M.d.S.S.); gnakazato@uel.br (G.N.); 4Postgraduate Studies in Bioprocess Engineering and Biotechnology, Federal University of Paraná (UFPR), Palotina 85950-000, PR, Brazil; greis.bio@gmail.com; 5School of Pharmaceutical Sciences, São Paulo State University (UNESP), Araraquara 14800-901, SP, Brazil; albertojuniorap@gmail.com (A.G.T.J.); victorhunterhsa@hotmail.com (V.H.S.A.); marlus.chorilli@unesp.br (M.C.); 6Laboratory for the Isolation and Transformation of Organic Molecules, Institute of Chemistry, University of Brasília (UnB), Brasilia 70910-900, DF, Brazil; lucantos4@gmail.com; 7Post-Graduate Program in Pharmaceuticals Sciences, Faculty of Health Sciences, University of Brasilia (UnB), Brasilia 70910-900, DF, Brazil; vitoriarpsilva@gmail.com; 8Institute of Science and Technology, Federal University of São Paulo (UNIFESP), São Jose dos Campos 12231-280, SP, Brazil; braga.hc@gmail.com (H.d.C.B.); d.tada@unifesp.br (D.B.T.); 9Faculty of Ceilandia, University of Brasília (UnB), Brasilia 72220-900, DF, Brazil

**Keywords:** Amazon, guarana, metallic nanoparticles, AgNPs, antibacterial, antifungal

## Abstract

The green synthesis of silver nanoparticles (AgNPs) can be developed using safe and environmentally friendly routes, can replace potentially toxic chemical methods, and can increase the scale of production. This study aimed to synthesize AgNPs from aqueous extracts of guarana (*Paullinia cupana*) leaves and flowers, collected in different seasons of the year, as a source of active biomolecules capable of reducing silver ions (Ag^+^) and promoting the stabilization of colloidal silver (Ag^0^). The plant aqueous extracts were characterized regarding their metabolic composition by liquid chromatography coupled to high-resolution mass spectrometry (UHPLC-HRMS/MS), phenolic compound content, and antioxidant potential against free radicals. The synthesized AgNPs were characterized by UV/Vis spectrophotometry, dynamic light scattering (DLS), nanoparticle tracking analysis (NTA), transmission electron microscopy (TEM), and scanning electron microscopy coupled to energy-dispersive X-ray spectrometry (EDX). The results demonstrated that the chemical characterization indicated the presence of secondary metabolites of many classes of compounds in the studied aqueous extracts studied, but alkaloids and flavonoids were predominant, which are widely recognized for their antioxidant capabilities. It was possible to notice subtle changes in the properties of the nanostructures depending on parameters such as seasonality and the part of the plant used, with the AgNPs showing surface plasmon resonance bands between 410 and 420 nm using the leaf extract and between 440 and 460 nm when prepared using the flower extract. Overall, the average hydrodynamic diameters of the AgNPs were similar among the samples (61.98 to 101.6 nm). Polydispersity index remained in the range of 0.2 to 0.4, indicating that colloidal stability did not change with storage time. Zeta potential was above −30 mV after one month of analysis, which is adequate for biological applications. TEM images showed AgNPs with diameters between 40.72 to 48.85 nm and particles of different morphologies. EDX indicated silver content by weight between 24.06 and 28.81%. The synthesized AgNPs exhibited antimicrobial efficacy against various pathogenic microorganisms of clinical and environmental interest, with MIC values between 2.12 and 21.25 µg/mL, which is close to those described for MBC values. Therefore, our results revealed the potential use of a native species of plant from Brazilian biodiversity combined with nanotechnology to produce antimicrobial agents.

## 1. Introduction

The relevance of silver (Ag) has been described for almost six millennia, being essential in several nations for various applications, from the construction of objects to its use in medicine [1]. There are records of the use of colloidal silver in the prevention of microbial infections, which was the most important agent before the introduction of antibiotics. In nanotechnology, silver can be used to synthesize nanoparticles with sizes between 1 and 100 nm, endowed with peculiar physical–chemical properties depending on both their size and morphology, enhancing their applications [2,3].

Silver nanoparticles (AgNPs) have high chemical stability and electrical conductivity, enabling their use in various functions, including optical, electronic, electromagnetic, and catalytic, and volumetric characteristics, such as chemical stability and electrical conductivity. They have a wide range of applications, such as in pharmaceutical formulations, photothermal therapy, drug delivery systems, electronic components, imaging and diagnostic agents, and photocatalytic degradation of dyes, among others [4,5]. Furthermore, due to the presence of the surface plasmon resonance (SPR) phenomenon, AgNPs have attracted unparalleled attention as color-based biosensors [6].

The antimicrobial properties of AgNPs stem from their ability to release silver cations (Ag^+^) from the surface [7]. Ag^+^ can interact with various cellular structures and processes in microorganisms, resulting in the inhibition of bacterial growth, disruption of cell wall integrity, and interference with essential metabolic processes. Additionally, AgNPs can penetrate the bacterial and fungal cells, causing direct damage to genetic material and leading to cell death [8]. Another known mechanism is the induction of oxidative stress by AgNPs on cells, which is neutralized by the defensive responses of these cells, including enzymatic and nonenzymatic defensive processes. When oxidative stress overwhelms these defense mechanisms, reactive oxygen species (ROS) and different types of free radicals damage the cell wall and biomolecules, including proteins, lipids, and DNA [9]. Additionally, due to their immediate effects on cell walls and their components, oxidative stresses can alter gene expression [3].

The conventional synthesis of AgNPs relies on the use of toxic solvents and energy-intensive equipment. In contrast, green synthesis is carried out using simple, scalable methodologies that replace toxic solvents with environmentally friendly raw materials, such as plant extracts, microorganisms, and enzymes that promote the cascade bioreduction of Ag^+^ ions, forming AgNPs. The presence of secondary metabolites with reducing activity in plant extracts, such as flavonoids, terpenoids, tannins, saponins, and phenolic compounds, is responsible for the formation of AgNPs. In addition, the deposition of biomolecules from the extract on the surface of the AgNPs increases their colloidal stability and can provide them further relevant biological activities for the development of therapeutic formulations. These advantages have encouraged several researchers on the use of plant extracts to synthesize AgNPs [4,10].

Brazilian biodiversity harbors many plants cataloged worldwide, positioning Brazil as a key source for bioeconomic development. Among native Brazilian, *P. cupana*, found in the northern Amazon biome, yields the guarana fruit, widely acclaimed for its neuro- and cardio-stimulating effects [11,12]. These effects are attributed to its phytochemical composition, notably high concentrations of methylxanthines such as caffeine, theobromine, and theophylline, as well as catechins, condensed tannins, polysaccharides, and proteins [12,13,14,15]. Furthermore, several studies have reported the antimicrobial properties of guarana against bacterial and fungal strains, including *Escherichia coli*, *Pseudomonas aeruginosa*, *Proteus vulgaris*, *Proteus mirabilis*, *Enterococcus faecalis*, *Staphylococcus aureus*, *Bacillus subtilis*, *Candida albicans*, *Botrytis cinerea*, and *Rhizoctonia solani* [11,16,17].

In this context, guarana can enhance the synthesis of AgNPs, affording the synergistic antimicrobial activity of silver and plant phytochemical compounds within a nanostructured system. Therefore, the objective of this study was to develop AgNPs through green synthesis, utilizing extracts of guarana leaves and flowers collected during different seasonal periods and to assess the antimicrobial efficacy of these nanoparticles against bacteria, fungi, and yeasts.

## 2. Results

### 2.1. Phytochemical Profile of Aqueous Extracts Characterized by UHPLC-HRMS/MS

The characterization of the phytochemical components in the *P. cupana* extracts using the UHPLC-HRMS/MS technique revealed the presence of various biomolecules, mostly from the alkaloids and flavonoids classes, as well as also phenolic acids, terpenoids, and procyanidins. Table 1 shows the data related to retention time and mass/charge ratio (*m*/*z*) from the positive and negative ionization modes, as well as the molecular formula and nomenclature of the biomolecules identified during the UHPLC-HRMS/MS experiments.

Evaluating the base peak chromatogram (BPC) of the leaf extract collected during the dry period (Ext-LD), metabolites 3, 4, and 7 showed protonated ions, with *m*/*z* ratios at 176.0919, 148.0976, and 130.0862 [M+H]^+^, suggesting the presence of calystegin B2, fagomine, and pipecolic acid, as well as peak 7 representing a deprotonated molecule compatible with quinic acid, with *m*/*z* at 191.0566 [M+H]^−^, and in all these cases the compounds were rapidly eluted, with a retention time (t_R_) of less than 2 min (Appendix A). Peaks 9, 11, 12, and 14 were also detected in this extract (Appendix A), representing the metabolites theobromine (181.0725; t_R_ 8.332 min), astilbin (449.1110; t_R_ 14.262 min), quercitrin (447.0950; t_R_ 14.933 min), and afzelin (431.0997; t_R_ 15.752 min), respectively.

The BPC of the extract of the leaves collected in the rainy season (Ext-LR) showed that peaks 6 and 7 [M+H]^+^ are the alkaloids calystegin B2 (*m*/*z* 176. 0918) and fagomine (*m*/*z* 148.0973), with t_R_ of 0.953 and 1.124 min, respectively (Appendix A), while peak 3 also represents the calystegin B2 molecule, but in the deprotonated state, with *m*/*z* 174.0768 [M+H]^−^ and t_R_ 0.979 min (Appendix A). In addition, the negative ionization mode revealed the presence of molecules derived from terpenoids, such as genipin (t_R_ 9.510 min; *m*/*z* 225.0760) and mikanolide (t_R_ 11.134 min; *m*/*z* 289.0705) for peaks 9 and 12, respectively, as well as a compound derived from procyanidins (t_R_ 10.559 min; *m*/*z* 577.1339) indicated in peak 11 (Appendix A).

In the aqueous extract of guarana flowers (Ext-FL), phytochemical investigation using BPC revealed a total of 10 compounds. As with the leaf extracts, the presence of the metabolites calystegin B2 (peak 4; t_R_ 1.030 min) and theobromine (peak 13; t_R_ 8.138 min) was detected in the positive ionization mode (Appendix A), and quinic acid (peak 4; t_R_ 1.218 min), mikanolide (peak 8; t_R_ 11.490 min), and quercitrin (peak 13; t_R_ 14.823 min) were identified in the negative ionization mode (Appendix A). In addition, it was possible to attribute the presence of the alkaloids trigonelline in peak 7 (t_R_ 1.261 min; *m*/*z* 138.0546) and caffeine in peak 15 (t_R_ 10.535 min; *m*/*z* 195.0880), both in the positive [M+H]^+^ ionization mode (Appendix A) and the flavonoids luteophorol (peak 17), cinnamtannin D1 (peak 19) and quercetin-*O*-malonylglucoside (peak 25) with retention times of over 11 min and precursor ions at *m*/*z* 291.0867, 865.2003, and 551.1049 [M+H]^+^ (Appendix A), respectively.

The results highlighted in Table 1 and Appendix A show that there is clear seasonal variation in the chemical composition when comparing the leaves’ aqueous extracts of *P. cupana* in the dry and rainy seasons. Similarly, distinct compounds are evident exclusively in the aqueous extract of *P. cupana* flowers. All these results observed are in accordance with the analysis of *P. cupana* seed extracts from different Brazilian states (Amazonas, Bahia, and Mato Grosso); the majority compounds and chemical markers were the molecules caffeine, catechin, epicatechin, and procyanidins [18]. Recently described was the presence of theobromine, caffeine, quercetin, procyanidins, among other major compounds in the extract of *P. cupana* flowers collected in the same geographical region and same month/year as the plant materials in the present study [19].

Given the diversity of phytochemical compounds available for green synthesis reactions of AgNPs, our study aligns with previous reports demonstrating that molecules such as caffeine, quercetin, and quercitrin from plant extracts play a key role in synthesizing of metallic nanostructures. Theses mechanisms involve keto-enol conversion (tautomerization) and the release of the reactive hydrogen atom that reacts with the dissociated metal and forms stable complexes [20,21,22,23]. In addition, terpenoids have been reported to act as reducing agents in the synthesis of AgNPs due to the inductive effect of their methoxy and allyl groups that remove free electrons from the *para* and *ortho* positions of hydroxyls, promoting the formation of stable resonant structures with subsequent stabilization of metal ions and the oxidation of aldehyde groups in carboxylic acids [24,25,26,27].

### 2.2. Quantification of Total Phenol Content (TPC) and Antioxidant Capacity of Aqueous Extracts of Paullinia cupana Leaves and Flowers

As demonstrated by the UHPLC-HRMS/MS technique, several biomolecules present in the *P. cupana* plant extracts are phenolic compounds which are antioxidant agents and participate in the bioreduction and stabilization of AgNPs through binding and chelation processes with silver ions (Ag^+^) [28,29,30]. Table 2 presents the results of the phenolic compound content and antioxidant potential of the plant extracts of *P. cupana* leaves and flowers. Additionally, the analytical curves constructed for the determination of each of these activities are provided in the Appendix A.

There was a variation in the content of phenolic compounds depending on the extract analyzed. Among the leaves, the TPC of Ext-LR (728.4 ± 0.087 µg GAE/g) was higher than that of Ext-LD (437.5 ± 0.093 µg GAE/g). Ext-LR also exhibited higher TPC compared Ext-FL (646.8 ± 0.165 µg GAE/g), demonstrating the influence of the seasonal period of collection of *P. cupana* and the type of the plant material on metabolite content. A previous study reported,119 mg GAE/g of phenolic compounds in the aqueous extract of guarana seed when using a boiling temperature of 100 °C [31]; however, in another study, a higher content of phenolic compounds (434 mg GAE/g) was detected from the hydroalcoholic extract of guarana seed prepared at 60 °C [32]. Generally, the content of phenolic compounds in plants depends on various aspects, including biotic and abiotic stresses, as well as the production and cultivation system employed, which can be modulated by the time of collection of the botanical plant materials [33,34].

Due to their redox characteristics, phenolic compounds possess high antioxidant potential. When adhered to the surface of AgNPs, they can increase this effect on nanostructures and promote long-term stability. Regarding the *P. cupana* plant extracts, there was great similarity in their DPPH and ABTS free radical scavenging activities, as the former ranged from 40.37 ± 0.008 to 41.72 ± 0.023 µg GAE/g and the latter from 19.24 ± 0.002 to 19.26 ± 0.002 µg GAE/g. This suggests that TPC had an impact on the antioxidant activity of guarana extracts and that the various biomolecules present in the plant’s phytochemical composition act synergistically to improve its biological activity [35].

### 2.3. Visual Aspects and UV/Vis Spectral Analysis

Visual inspection at the end of the synthesis showed that the optical characteristics of the AgNPs were similar, with color intensity not varying, depending on the part of the plant used and the seasonal period (Appendix A). In turn, the kinetic curves of the AgNPs show an increase in absorbance intensity for all the samples at 30 min of the reaction at 450 nm, while no change was seen in the control samples (Figure 1A). The results show that AgNPs-LR reached the highest intensity after 180 min of incubation (0.858 a.u.) compared to AgNPs from the dry period, which had similar absorbance intensity after the end of the reactions (0.630 a.u. for AgNPs-LD and 0.617 a.u. for AgNPs-FL). The progressive increase in the synthesis rate of AgNPs, proportional to the reaction time, may indicate that the process of reducing silver (Ag^+^) into colloidal silver (Ag^0^) is taking place together with the AgNPs growth stage [36].

Figure 1B shows the UV/Vis electronic spectra of the AgNPs monitoring the bands of maximum absorption after synthesis. Overall, the AgNPs showed a progressive increase in absorbance intensity during the 30 days of spectral analysis carried out, accompanied by the formation of more stable nuclei around the AgNPs [37]. The maximum absorbances were between 410 and 420 nm for AgNPs-LD and AgNPs-LR, while for AgNPs-FL the absorption bands were considerably broader and with wavelengths between 440 and 460 nm. These differences in the SPR bands and the redshifts may be due to changes in the sizes and/or shapes of the biogenic AgNPs [38]. The alteration in color of the reaction mixtures and the absorptions at different wavelengths may be attributed to the surface plasmon resonance (SPR) phenomenon. This phenomenon arises from the interaction of light with the nanoparticulate system, leading to oscillations of free electrons on the surface of the AgNPs [39].

In addition, when using the *P. cupana* leaf extract, the absorption curves of the AgNPs showed higher relative intensities compared to using the flower extract, as seen previously in a study that synthesized AgNPs from different plant parts of *Handroanthus heptaphyllus* [40]. This effect may be related to the chemical composition of the leaves, which contain high concentration of biomolecules [41], as well as suggesting a better yield in the synthesis of AgNPs given the increase in the intensity of the SPR bands [42].

### 2.4. Evaluation of Colloidal Stability by DLS and Zeta Potential

The dimensional properties of the AgNPs revealed an average hydrodynamic diameter of 79.29 ± 17.4 nm for the AgNPs-LD, which was 8.61% smaller than of the measurement obtained for the AgNPs-LR on the day of synthesis, which was 86.76 ± 9.9 nm. In the assessment of colloidal stability, none of these groups of samples showed any statistically significant differences over one month of analysis. On the other hand, the AgNPs-FL showed significant changes (*p* < 0.05) in the average hydrodynamic diameter, given that after one week the aliquots at room temperature and under refrigeration had a decrease of 19.28 and 13.1%, respectively, as well as the sample stored at room temperature after one month, which exhibited a reduction of 15.56% in its diameter compared to the initial measurement of 78.87 ± 4.1 nm (Table 3).

The PdI represents the size distribution of nanostructures, and its values above 0.5 indicate a heterogeneous size range [43]. When checking the values for AgNPs-LD and AgNPs-FL, both synthesized with extracts of plant parts from the dry period, no significant differences were observed after thirty days of storage, varying between 0.277 ± 0.008 and 0.479 ± 0.062. However, the PdI of the AgNPs-LR showed statistical differences (*p* < 0.05) about the initial values, showing decreases of 25.42% (D1-REF), 22.78% (D7-RT), and 28.30% (D30-RT), which demonstrates the improvement in the homogeneity of the particles over time and less possibility of aggregation (Table 3).

Zeta potential was the parameter that varied the most during the month of monitoring. Statistically significant differences (*p* < 0.05) were observed in the AgNPs-LD after seven and thirty days in both storage conditions, decreasing by 35.9% and between 27.52 and 36.24%, respectively, compared to the initial measurement of −28.7 ± 0.80 mV, which represents moderate stability of the nanoparticulate system [44]. On the other hand, in the case of AgNPs-FL and AgNPs-LR, most of the values obtained over time were above −30 mV (moderate stability), varying significantly (*p* < 0.05) only after one day at room temperature, with a reduction of 21.3% compared to the initial measurement and 56.36% after one month of storage under refrigeration, which can be indicated as probable colloidal instability (Table 3).

There is a relative lack of studies evaluating the variables described here, such as the effect of seasonality on the collection of biological material used in the green synthesis of AgNPs and the comparison of colloidal characteristics when using different parts of the same plant or even the same botanical material from different plant species. A study carried out with AgNPs synthesized from extracts of *Pterodon emarginatus* leaves collected in different seasonal periods showed that the values related to average hydrodynamic diameter, polydispersity index, and zeta potential did not vary even after five months of storage [45]. In another study, the average size of AgNPs synthesized from the leaves of *Aloe vera*, *Coriandrum sativum,* and *Cymbopogon citratus* had a distribution between 10 and 22 nm, 5 and 37 nm, and 10 and 50 nm, respectively [46]. In our previous study, the AgNPs synthesized from extracts of the leaves of three plants from the Arecaceae family showed variation in average size (130.43–352.93 nm), with high PdI (0.523–0.689) and zeta potential with incipient instability (−17.2 to −26.97 mV), demonstrating that even with similar phytochemical compositions, plant extracts can give significantly different results, compromising the reproducibility of syntheses [47].

It is important to note that the differences in the characteristics observed during the stability tests of AgNPs from *P. cupana* may be related to (i) the time when the plant material was collected/located, (ii) the part of the plant used, and (iii) the temperature/environment in which the colloidal suspensions were stored. Together, these factors can influence the availability of the biomolecules to the Ag^+^ reduction processes and can also affect the interaction of the covering agents with the surface of the AgNPs, modifying both size and homogeneity of the colloidal system [18,48,49]. In this way, such evaluations are of fundamental importance when monitoring the physicochemical parameters of biogenic nanostructures even in syntheses carried out under similar conditions, where it is still expected that such variations can occur.

### 2.5. Nanoparticle Tracking Analysis (NTA)

NTA analyses make it possible to calculate the average diameter and concentration of the particles quickly and easily. In our study, the measurements revealed that the sizes of the nanostructures synthesized with leaf extracts were 68.9 ± 0.7 nm for AgNPs-LD and 78.4 ± 2.6 nm for AgNPs-LR, with suspensions containing 1.56 × 10^8^ and 1.43 × 10^10^ particles/mL, respectively. In turn, the highest concentration of particles was observed in AgNPs-FL, with 1.68 × 10^11^ particles/mL and a hydrodynamic diameter of 61.4 ± 1.0 nm (Table 4).

In general, the hydrodynamic diameters measured by NTA were similar to those found by DLS analyses throughout the monitoring of colloidal stability, as described in Table 3. In addition, it was evident that the concentration of particles was higher in the sample synthesized with the flowers (AgNPs-FL), corroborating previous studies in which a higher concentration of secondary metabolites, mainly alkaloids and flavonoids, were found in this organ [50]. This may be due to the higher concentration of alkaloid transport proteins in the reproductive part of plants, which also facilitates their reproductive function, increases the concentration of biomolecules in these organs, and, from their reducing and stabilizing capacity, these compounds can participate in the green synthesis process of AgNPs [51].

The use of this technique to characterize nanostructures from various plants has previously been described in studies that synthesized AgNPs from extracts of *Azadirachta indica* leaves and the mixture of herbs called Triphala, with diameters of 43 and 59 nm, respectively, as well as the *Elephantopus scaber* leaf extract that resulted in AgNPs with 78 nm and a concentration of 1.66 × 10^8^ particles/mL [52,53]. These characteristics, intrinsic to each group of particles, can be correlated to the chelating and reducing capacity of the phytochemical compounds available in the plant extracts, which can generate different properties in the synthesized AgNPs.

### 2.6. Transmission Electron Microscopy (TEM) and Energy-Dispersive X-ray Spectroscopy (EDX)

The AgNPs characterized by TEM showed regular, well-defined edges and a predominantly spheroidal morphology. It was also possible to observe triangular, hexagonal, prismatic, and anisotropic shapes, which may be associated with the different ways in which the particles nucleate because of the overly complex organic environment (Figure 2). Corroborating these results and based on Mie’s theory, it has been reported that the presence of only a single maximum absorption band in UV/Vis spectra is considered characteristic of spherical nanostructures and, when the UV/Vis spectra depict maximum of absorption at different wavelengths, the suspension may contain AgNPs of different shapes and sizes [54].

The average dry size for AgNPs-LD, AgNPs-FL, and AgNPs-LR was 46.37 ± 0.18 nm, 48.85 ± 0.19 nm, and 40.72 ± 0.16 nm, respectively (Figure 2), making it possible to infer that the probable changes in the phytochemical profiles of the aqueous extracts did not strongly modify the dry diameter of the AgNPs. Our results are in line with previous studies where AgNPs measuring 43 nm were synthesized using the aqueous extract of *Cassia auriculata* flowers prepared by heating [55] and another report that investigated the biosynthesis of AgNPs from *Stachys inflata* leaf extract with a size in the 35–45 nm range [56]. In turn, when *Pterodon emarginatus* leaves collected in the dry and rainy seasons were used for AgNPs biosynthesis, the average diameters obtained were 28.1 nm and 33.2 nm, respectively [45]. In addition, different plant organs from the same plant (*Ficus deltoidea*) were used in the green synthesis of AgNPs and it was evident that the average diameter of the nanostructures was similar, ranging from 15.7 nm (root) to 20.5 nm (stem) and 22.9 nm (leaves) [57].

The results show different parts of the plant, and the extraction method can affect both the morphology and the dimensions of the AgNPs. In addition, geographical variations can lead to inconsistent morphological properties based on variations in the profile of metabolites in the extracts used in phytosynthesis [58]. Another important aspect is the layer of coating covering the surface of the nanostructures, giving them a shaded appearance. This phenomenon is characteristic when analyzing metallic nanostructures synthesized by biogenic routes due to the immobilization of biomolecules from plant extracts that act as stabilizers during the synthesis process [59,60,61].

The elemental compositions of the AgNPs samples were assessed by the EDX technique based on the interaction of X-rays with the samples that was coupled to a scanning electron microscope. The images next to each spectrum represent the overlay mapping of all the elements present in the samples and are presented according to their spatial distribution (Figure 3). The analyses show an absorption signal around 3 keV which is characteristic of the presence of silver which is dispersed at the beginning of the spectra as one of the main constituents [62]. In the spectrum of AgNPs-LD, silver has a mass proportion of 24.06%, while for AgNPs-FL and AgNPs-LR this percentage increases to 28.81 and 25.91%, respectively, agreeing with recent studies that reported silver to be one of the main surface elements [63,64,65].

In addition, the EDX spectra revealed the presence of peaks with the predominant identification of carbon (C) and oxygen (O) atoms, as well as magnesium (Mg), silicon (Si), and potassium (K), which may be related to the remnants of the biomolecules previously identified in the aqueous extracts of *P. cupana* leaves and flowers, as well as appearing as traces of the mineral content coming from the soil in which the plant is grown [66,67,68]. The intense peak around 1.5 keV indicates the presence of aluminum which makes up the support on which the samples were previously deposited for reading in the equipment.

### 2.7. Evaluation of Antimicrobial Activity

The antibacterial activity of AgNPs-LD and AgNP-FL showed that the minimum inhibitory concentrations (MICs) against the Gram-negative strain *Escherichia coli* were 2.65 and 5.3 µg/mL, respectively, while for AgNPs-LR the lowest concentration capable of inhibiting bacterial growth was 10.6 µg/mL. Overall, these values were the same as those reported by minimum bactericidal concentration (MBC), and lower than the values described for the aqueous AgNO_3_ solution (positive control) (Table 5). In addition, our results corroborate recent research that showed an MIC of 2.5 µg/mL for AgNPs produced from *Ficus deltoidea* leaf extract, as well as another study in which AgNPs synthesized from *Rotheca serrata* flower extract had an MIC of 12 µg/mL for *Escherichia coli* [57,69].

When looking at the results with the Gram-positive bacterium *Staphylococcus aureus*, regardless of the type of nanostructure tested, the MIC was 10.6 µg/mL, with the same effect caused by the aqueous AgNO_3_ solution and with values equal to or twice as high as the MBC values, showing that this bacterium was less susceptible to the nanostructures (Table 5). In other studies, with this same bacterium, AgNPs synthesized with *Jatropha integerrima* flower extract had an MIC of 5 μg/mL [70], while AgNPs from *Teucrium polium* leaf extract had an MIC of 25 µg/mL [71].

The fungistatic effect of AgNPs was observed for all the species tested, with the MIC values standing out for the fungi *Aspergillus fumigatus* and *Penicillium chrysogenum*, which were sensitive to concentrations of less than 3 µg/mL regardless of the sample tested (Table 5). Other studies with these same fungal species have shown the formation of growth inhibition halos after treatment with different concentrations of biogenic AgNPs synthesized from *Aquilegia pubiflora* and *Mussaenda glabrata* extracts [72,73].

In turn, the yeast *Candida albicans* exhibited greater sensitivity to samples from the dry period compared to those from the rainy period, with an MIC of 2.12 µg/mL after treatment with AgNPs-LD and 5.3 µg/mL for AgNPs-LR. This behavior was the same as that described for the fungus *Fusarium oxysporum*, which showed MICs between 4.25 and 21.25 µg/mL (Table 5). These results are more promising than data reported in other studies in which AgNPs synthesized from the fruit extract of *Scabiosa atropurpurea* had an MIC of 7.81 µg/mL [74] and in a study in which the antifungal activity of AgNPs obtained from the leaf extract of *Malva parviflora* showed an 80.7% reduction in the mycelial growth of the *Candida albicans* species [75].

Previous studies have reported greater susceptibility of Gram-negative bacteria to AgNPs due to their cell wall being composed of a single layer of peptidoglycan, making it less rigid, as well as the presence of negatively charged lipopolysaccharides which, together, promote the attraction of silver ions and favor the entry of agents such as nanometer-scale particles through specific structures such as porins. On the other hand, the thicker cell wall of Gram-positive bacteria protects against the penetration of nanostructures into the cytoplasm [76,77,78].

Regarding the mechanisms of action of AgNPs on fungal activity, changes in the permeability of the plasma membrane have already been reported, with modifications to its integrity and polarization. In addition, the high reactivity of the particles can trigger changes in mitochondrial functionality, reducing ATP synthesis, causing fragmentation of genetic material, and reducing gene transcription [79,80]. Microscopic analyses have also shown shortening and condensation of hyphae, cell plasmolysis, disintegration of organelles, and changes in the number of germinated conidia after treatment with AgNPs [81,82].

In common, some mechanisms of antimicrobial action of biogenic AgNPs indicate (i) size as an important parameter that allows nanostructures to be internalized into microbial cells, (ii) electrostatic interaction with chemical groups present in the microbial cell membrane leading to dissipation of the proton motive force and cell death, (iii) the production of reactive oxygen species (ROS) and free radicals that cause oxidative stress and can inhibit cellular respiration, and (iv) binding to thiol and sulfhydryl groups at the genomic level, which inactivates enzymes vital for cell replication and metabolism [83,84,85,86]. In this way, this may indicate that the AgNPs synthesized in this study become ideal candidates for applications in various areas, such as the food and health industries, fighting microorganisms that cause infections and are resistant to the drugs used in clinical practice.

The aqueous extracts of *P. cupana* had no antimicrobial action within the range of concentrations tested, and, as far as we know, this was the first investigation of such biological activity using the leaves and flowers of this plant. Some factors, such as the amount of extract used in the synthesis, the extraction method, and the solvent used during the extraction of the biomolecules, as well as the period when the plant material was collected (seasonality), may have influenced these results. However, guarana seed extracts showed antibacterial efficacy against *Escherichia coli* and *Staphylococcus aureus* with MICs of 32 and 64 µg/mL, respectively [87], as well as in another study where a slight inhibition of *Staphylococcus aureus* growth was reported after treatment using guarana seed extract [88]. Antifungal activity against the yeast *Candida albicans* using guarana seed extracts prepared with organic solvents revealed MICs between 250 and 500 µg/mL [11]. In another study, fungal species of the genus *Aspergillus* and *Penicillium* showed growth inhibition of between 13.4 and 64.4% after being exposed to the aqueous extract of guarana seeds [31].

## 3. Material and Methods

### 3.1. Chemicals and Reagents

In this study all chemicals were used without any further purification. Silver nitrate (AgNO_3_) (Sigma-Aldrich, St. Louis, MO, USA), formic acid (CH_₂_O_₂_), and acetonitrile (C_₂_H_₃_N) were of LC-MS analytical grade (>98%) obtained from Sigma Aldrich (St. Louis, MO, USA). DPPH and ABTS free radicals were purchased from Sigma-Aldrich (St. Louis, MO, USA). Methanol (CH_₃_OH) (Dinâmica, Indaiatuba, São Paulo, Brazil), gallic acid (C_7_H_6_O_5_) (Dinâmica, Indaiatuba, São Paulo, Brazil), and Folin–Ciocalteu (Dinâmica, Indaiatuba, São Paulo, Brazil) were obtained from the same manufacturer.

The microorganisms *Escherichia coli* (ATCC 25922), *Staphylococcus aureus* (ATCC 25923), and *Candida albicans* (ATCC 90028) were obtained from the *American Type Culture Collection* (ATCC). The fungi *Aspergillus fumigatus*, *Fusarium oxysporum,* and *Penicillium chrysogenum* were isolated from the environment and kept in the laboratory. The growth media Muller–Hinton Agar and Sabouraud Dextrose Agar were obtained from Difco (Sigma Aldrich, St. Louis, MO, USA) and Kasvi (Pinhais, Paraná, Brazil), respectively. Deionized water (DI) was used to prepare the plant extracts, the metal salt solutions, and for other purposes during the research.

### 3.2. Collection of Plant Material and Preparation of Aqueous Extracts of Paullinia cupana Leaves and Flowers

The botanical materials of *P. cupana* were collected from a private property—Sitio Putitanga 03°22′07″ S and 57°41′27″ O—located in the city of Maues in the state of Amazonas, Brazil. All procedures were authorized by the Genetic Heritage Management Council (CGEN) under regularization number A5C4D66. The leaves were collected in August and February, which correspond to the winter and summer seasons or the dry and rainy periods in the Amazon, respectively. The flowers were collected according to spontaneous demand and therefore only in August (dry season). It is worth noting that all the collections were made early in the morning from the same plant. When they arrived at the laboratory, the plant materials were stored at −80 °C until they were used.

To prepare the aqueous extracts, the leaves and flowers were thawed at room temperature and then washed with plenty of distilled water, then with neutral detergent, and again with distilled water to remove impurities from their surfaces. After drying with a paper towel, 1 g of each plant material was cut into fragments of a similar size (5 mm^2^) and added separately to a beaker containing 10 mL of deionized water and left to boil on a hotplate (RH Basic 2, IKA) for 3 min. Afterwards, each extract was filtered through qualitative filter paper (Whatman nº1) and the concentration of each preparation was 100 mg/mL (*m*/*v*). The extracts obtained were labeled Ext-LD (aqueous extract of leaves collected during the dry season), Ext-FL (aqueous extract of flowers), and Ext-LR (aqueous extract of leaves collected during the rainy season).

### 3.3. Phytochemical Characterization and Antioxidant Potential of Paullinia cupana Plant Extracts

#### 3.3.1. UHPLC-HRMS/MS

The composition of plant metabolites in *P. cupana* leaf and flower extracts was putatively assigned using UHPLC-HRMS/MS. Briefly, the plant extracts were diluted to a concentration of 2 mg/mL in deionized water, centrifuged at 13,200 rpm for 10 min, and a flow rate of 0.4 mL/min and 1 µL as the injection volume was used for the analyses. The chromatographic run conditions and equipment settings, as well as analysis methods and databases consulted to confirm the results obtained, are described in our previous study [89].

#### 3.3.2. Quantification of Total Phenol Content (TPC)

TPC was determined using the Folin–Ciocalteau method [90]. In 96-well microplates, 20 µL of the aqueous extracts (100 mg/mL) were mixed with the Folin–Ciocalteau reagent (Dinâmica, Brazil), and then 60 µL of aqueous sodium carbonate solution (10%) was added per well and incubated for 20 min in the dark. After this period, the phenolic content was measured by measuring the absorbance of the wells in a spectrophotometer combined with a microplate reader (SpectraMax M3, Molecular Devices, São Paulo, Brazil) at 760 nm. The standard curve was prepared using methanolic solutions of gallic acid (GA) at various concentrations, and the phenolic content was expressed as equivalents of the standard per mass of plant extract (µg GAE/g). Three independent experiments were conducted, each in triplicate per concentration of the standard/sample, for both the standard curve and the quantification tests total phenol content.

#### 3.3.3. DPPH

The antioxidant potential of the aqueous extracts was assessed using the DPPH (2,2-diphenyl-1-picrylhydrazyl) free radical scavenging method (Sigma-Aldrich, Brazil), according to the methodology adapted [91]. Briefly, in 96-well microplates, 20 µL of the extracts (100 mg/mL) were mixed with 280 µL of a freshly prepared methanolic solution of DPPH (0.08 mM). The mixtures were then left to stand for 30 min, protected from light, at room temperature, and then the absorbance of the wells was read in a spectrophotometer conjugated to a microplate reader (M3, Molecular Devices) at 517 nm. The standard curve was constructed from methanolic solutions of gallic acid at different concentrations and the results were expressed as equivalents of the standard per mass of plant extract (µg GAE/g). All the tests were carried out in three independent experiments in triplicate per concentration of standard/sample.

#### 3.3.4. ABTS

The antioxidant capacity of the aqueous extracts was also tested by eliminating the cation radical ABTS^+^ (2,2-azino-bis(3-ethylbenzo-thiazoline-6-sophonic acid) (Sigma-Aldrich, Brazil), according to the previous study [92]. To obtain the radical, 2 mL of ABTS (7 mM) was mixed with 0.0352 mL of potassium persulfate (140 mM) and kept in the dark for around 16 h at room temperature. After this period, the ABTS^+^ produced was diluted in methanol until it reached an absorbance between 0.8 and 1 at 734 nm. Then, using a 96-well microplate, 20 µL of each extract (100 mg/mL) was mixed with 280 µL of the methanolic ABTS^+^ solution and incubated for 20 min at room temperature, with the absorbance of the wells subsequently read at 734 nm on a spectrophotometer conjugated to a microplate reader (M3, Molecular Devices). The standard curve was obtained from solutions of gallic acid in methanol at different concentrations, and the results were expressed as equivalents of the standard per mass of plant extract (µg GAE/g). All the tests were carried out in three independent experiments in triplicate per standard/sample concentration.

### 3.4. Green Synthesis of AgNPs Using Paullinia cupana Extracts

For the green synthesis of AgNPs with aqueous extracts of *P. cupana* plant parts, aqueous solutions of silver nitrate (AgNO_3_) at a final concentration of 1 mM (approximately 170 µg/mL) were added to glass test tubes and then the extracts were added in a volume equivalent to the concentration of 1 mg/mL in relation to the final volume of 6 mL, being 5.94 mL of the AgNO_3_ solution and 0.06 mL of the plant extract. Experimental controls were also prepared, consisting of (i) extract control (deionized water and the extract at a concentration of 1 mg/mL) and (ii) AgNO3 control (1 mM metal salt and deionized water), both in a final volume of 6 mL. The reaction mixtures were then protected with aluminum foil, to prevent photo-oxidation of the silver and evaporation of the liquid during the reaction, and incubated under cover of light in a water bath (555, Fisatom, São Paulo, Brazil) at 70 °C for 180 min. The colloidal suspensions were named AgNPs-LD (nanostructures synthesized from leaf extract collected during the dry season), AgNPs-FL (nanostructures synthesized from flower extract), and AgNPs-LR (nanostructures synthesized from leaf extract collected during the rainy season).

### 3.5. UV/Vis Spectrophotometric Analysis

The kinetics of the synthesis reactions were monitored every 30 min using 2 mL of each sample, undiluted, using a UV/Vis spectrophotometer (UV1800PC, Phenix, Blomberg, Germany), adjusting the wavelength to 450 nm during the incubation period of the colloidal suspensions and experimental controls. In addition, spectrophotometric analyses in the 350 to 500 nm range were conducted at the end of the reactions and seven and thirty days after the initial synthesis to explore the maximum absorption bands of each AgNPs colloidal suspension.

### 3.6. Dynamic Light Scattering (DLS) and Electrophoretic Mobility (Zeta Potential) Analysis

The AgNPs were analyzed for their average hydrodynamic diameter (DH), polydispersity index (PdI), and zeta potential (ZP) using the ZetaSizer Nano ZS equipament (Malvern Instruments, Malvern, UK). One mL of each sample diluted in a 1:10 (*v*/*v*) ratio, with 100 µL of AgNPs and 900 µL of deionized water, was placed in the polystyrene cuvette, which was inserted into the equipment configured as follows: stabilization time of 120 s before measurements, temperature of 25 °C, scattering angle of 90°, helium–neon laser operating at 633 nm, and 10 readings in triplicate.

Identical analyses were carried out one, seven, and thirty days after the initial synthesis to monitor the colloidal stability and physicochemical characteristics of the AgNPs over time. In addition, to evaluate the effect of storage, aliquots of each sample were separated and kept at room temperature (22 °C) and in a refrigerator (4 °C). The results were processed using the Zetasizer 7.13 software from the same equipment manufacturer.

### 3.7. Nanoparticle Tracking Analysis (NTA)

The NTA technique was employed to determine the hydrodynamic diameter and concentration of the AgNPs being carried out according to the methodology adapted [93]. To investigate these characteristics, the samples were first diluted in a 1:1000 (*v*/*v*) ratio in deionized water and injected with sterile syringes into the sample chamber and then analyzed using the Nano-Sight LM-10 HS System (NanoSight Ltd., Minton Park, Amesbury, Wiltshire SP47RT, UK), with the measurements being made in a single manual shutter for 60 s with gain, brightness, and threshold adjustments at room temperature, with a 532 nm (green) diode laser.

### 3.8. Transmission Electron Microscopy (TEM)

The AgNPs were analyzed by MET to determine their morphology and dry diameter. To perform this, 10 µL of the AgNPs, without prior dilution, were dripped onto a copper screen covered with Formvar^®^ film deposited on filter paper inside a glass Petri dish. After dripping, the samples were dried in an oven at 60 °C for 1 h and then analyzed using a transmission electron microscope (FEI, Tecnai, Hillsboro, OR, USA) operating at 120 kV. The photomicrographs were captured randomly, and the size distribution histograms were generated using OriginPro 8.5 software (OriginLab Corporation, Northampton, MA, USA) based on the particle count of each group of AgNPs with the aid of ImageJ software version 1.8.0 (National Institute of Health, Bethesda, MD, USA).

### 3.9. Energy-Dispersive X-rays (EDX)

To determine the elemental composition of the atoms in the samples, as well as to ascertain the silver content in each of the AgNPs suspensions, EDX analyses were carried out on a scanning electron microscope (FEI, INSPECT S50 with Everhart-Thornley detector, Hillsboro, OR, USA) operating at 10 kV. The AgNPs were deposited on aluminum supports (stubs) specific for the microscope to be used, without metallization, remaining for three days in containers protected from light until completely dry for subsequent analysis.

### 3.10. Antimicrobial Activity

The biological activity of the AgNPs, the aqueous extracts, and the AgNO_3_ solution were tested against different strains of bacteria and fungi by means of minimum inhibitory concentration (MIC) evaluations and, for bacteria, using the minimum bactericidal concentration (MBC).

Using the broth microdilution method, according to the Clinical Laboratory Standard Institute [94], suspensions of Gram-negative *Escherichia coli* (ATCC 25922) and Gram-positive *Staphylococcus aureus* (ATCC 25923) bacteria were sown in Petri dishes containing Muller–Hinton Agar (MH) culture medium (Difco, Franklin Lakes, NJ, USA), and incubated for 24 h at 37 °C. After growth, bacterial colonies of each strain were collected from the plates and placed in saline solution (0.85% NaCl) to adjust the optical density to 0.5 on the McFarland scale or 1.5 × 10^8^ CFU/mL. Then, 50 µL of MH was added to 96-well microplates, followed by 50 µL of each sample in the first well of the plate, totaling 100 µL. Briefly, 50 µL of this solution was removed and placed in the next well, and so on in all the wells per sample, carrying out a serial dilution with an initial concentration of 42.5 up to a concentration of 0.33 µg/mL of the AgNPs and AgNO_3_ and between 25,000 and 50 µg/mL of the aqueous extracts. To complete the final volume of 100 µL per well, 50 µL of saline solution was added with the bacterial suspension diluted to a concentration of 1 × 10^6^ UFC/mL in MH. The plates were incubated at 37 °C for 18 to 24 h, and then read visually, evaluated according to the turbidity of the medium, where the absence of turbidity indicated inhibition of the microorganism’s growth. MBC was determined from bacterial death ≥99.9% after 24 h of treatment.

The antifungal tests were carried out against the filamentous fungi *Aspergillus fumigatus*, *Fusarium oxysporum*, and *Penicillium chrysogenum* isolated from the environment, following document M38-A2, and against the yeast *Candida albicans* (ATCC 90028), based on the recommendations of document M27-A2 [95,96], grown on Sabouraud Dextrose Agar (Kasvi). The fungal suspensions were prepared in sterile phosphate-buffered saline (PBS), and after counting in a hematocytometer chamber, the inoculum was adjusted to 10^4^ CFU/mL for filamentous fungi and 10^3^ CFU/mL for yeast in RPMI 1640 medium plus MOPS buffer at pH 7.2 to 7.4. In 96-well microplates, 100 µL aliquots of each fungal suspension were deposited in the previously identified wells and then 100 µL of different concentrations of AgNPs and AgNO_3_ (42.5–0.02 µg/mL) and plant extracts (25,000–50 µg/mL) obtained through serial dilution were added to each well per concentration/sample. The microplates were incubated for 48 h at 30°C and then a visual reading was taken, where growth inhibition was assessed by the absence of turbidity in the culture medium.

All the biological tests were carried out in triplicate per concentration/sample, with the positive control consisting only of the microorganism (bacteria or fungi) and the negative control containing only the culture medium.

### 3.11. Statistical Analysis

The quantitative data regarding the total phenolic content and antioxidant potential of the extracts, as well as the physicochemical characterization data of the AgNPs obtained by DLS and the mean values of the dry diameter determination of the AgNPs obtained by TEM, are presented as mean ± standard deviation (SD) of the mean. Possible differences between the groups were determined by analysis of variance (ANOVA) followed by Tukey’s test with a significance level of *p* < 0.05. Graphs and/or histograms were plotted using GraphPrism 8 (GraphPad Software, San Diego, CA, USA) and OriginPro 8.5 (OriginLab Corporation, Northampton, MA, USA) software.

## 4. Conclusions

The green synthesis of AgNPs was performed, using a sustainable, simple, fast, and environmentally friendly approach employing aqueous extracts from the leaves and flowers of *P. cupana*, a plant native to Brazilian biodiversity. The temperature-controlled extraction method of plant biomolecules favored the elucidation of alkaloids, flavonoids, terpenoids, procyanidins, and phenolic acids, which play a direct role in the bioreduction and stabilization of nanostructures through different mechanisms that have not yet been unclear. Furthermore, seasonal variations and the plant tissue utilized directly correlate with the yield of AgNPs, as well as their hydrodynamic diameters, polydispersity, concentration, morphology, and zeta potential, demonstrating that these characteristics tailor nanostructures for specific applications. In addition, the AgNPs exhibited significant antimicrobial action against various bacterial and fungal strains, surpassing, in some cases, that of the positive control (aqueous AgNO_3_ solution) and regardless of the time of the plant material collection and the part of the plant.

## Figures and Tables

**Figure 1 pharmaceuticals-17-00869-f001:**
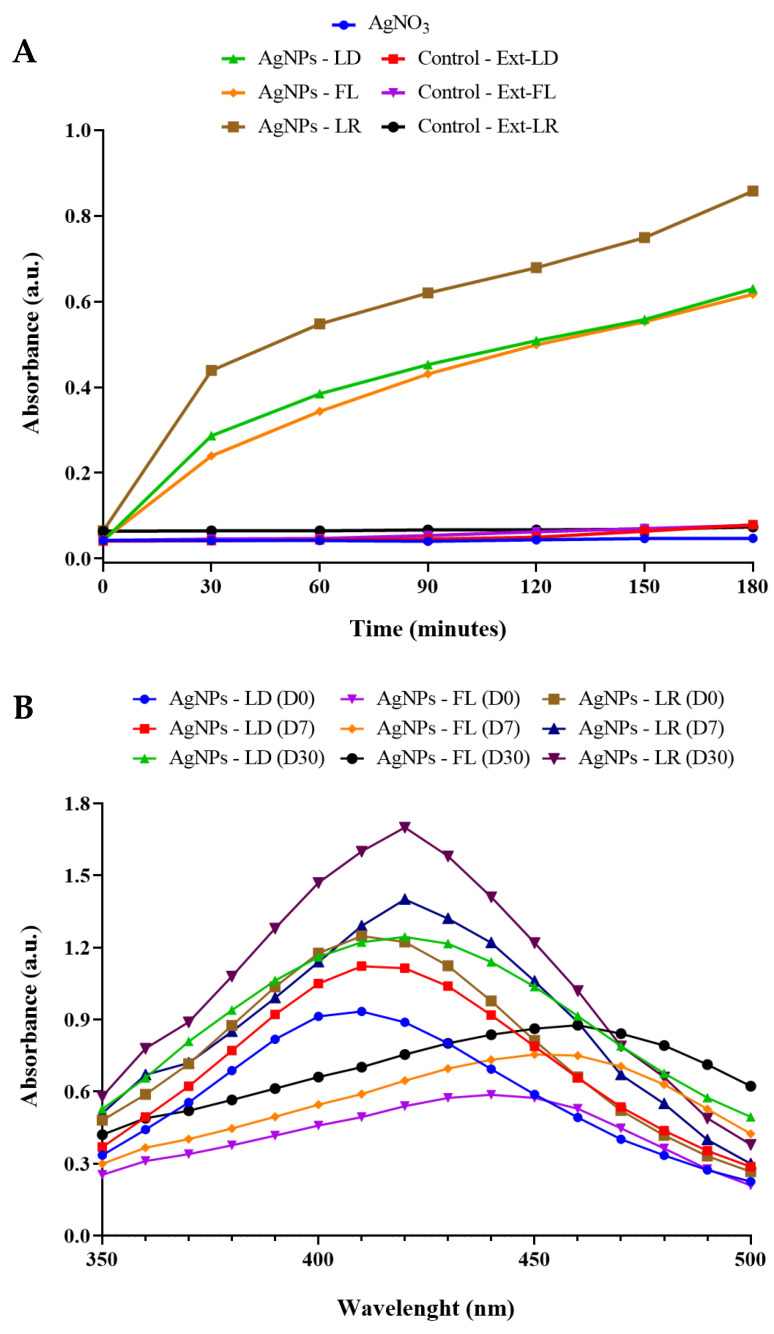
(**A**) Kinetic monitoring of AgNPs synthesis by spectrophotometric analysis at 450 nm for 180 min. (**B**) Absorption curves in the UV/Vis region of AgNPs in the range between 350 and 500 nm.

**Figure 2 pharmaceuticals-17-00869-f002:**
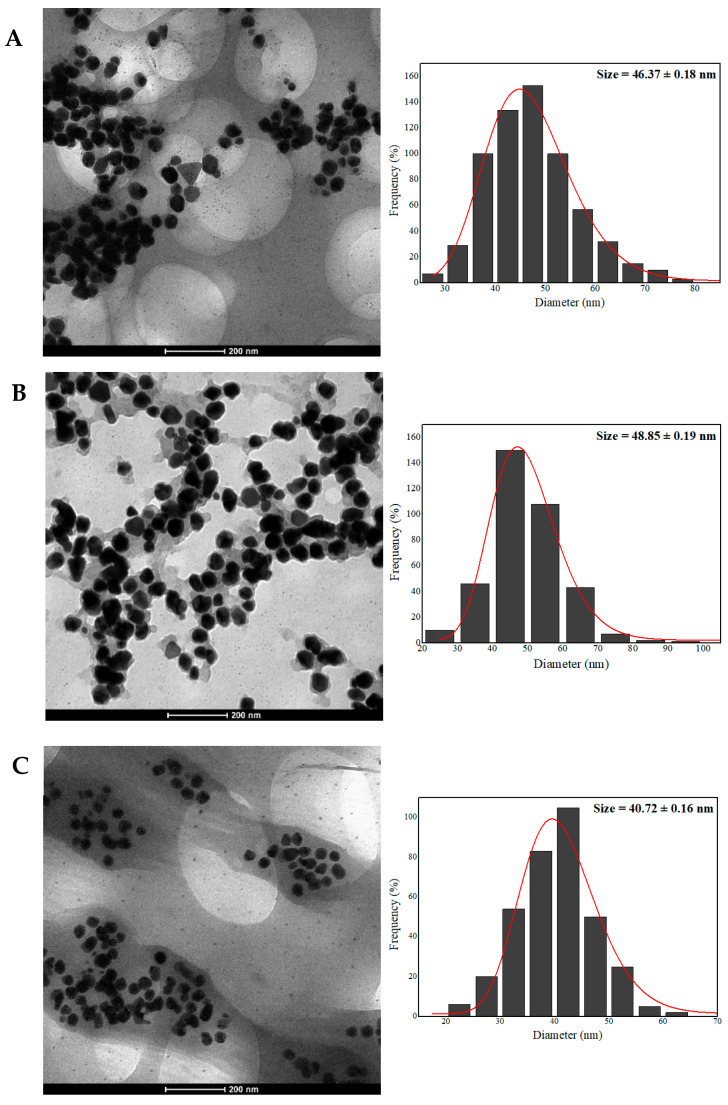
TEM micrographs and diameter distribution histograms of AgNPs-LD (**A**), AgNPs-FL (**B**), and AgNPs-LR (**C**).

**Figure 3 pharmaceuticals-17-00869-f003:**
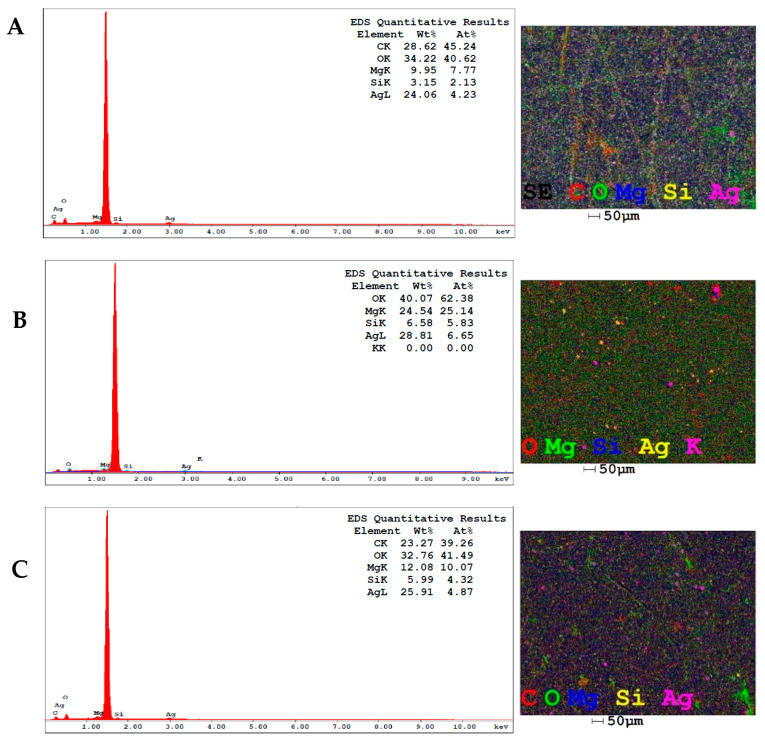
EDX spectra and elemental map showing the elemental profile in the AgNPs-LD (**A**), AgNPs-FL (**B**), and AgNPs-LR (**C**) samples.

**Table 1 pharmaceuticals-17-00869-t001:** Compounds identified by UHPLC-HRMS/MS in extracts of *Paullinia cupana* leaves collected during the dry season (Ext-LD), rainy season (Ext-LR), and flower extract (Ext-FL) collected during the dry season.

Ext-LD					
Peak	Retention Time (min)	*m*/*z* [M+H]^+^	*m*/*z* [M+H]^−^	Molecular Formula	Compound
3	1.000	176.0919	–	C_7_H_13_NO_4_	Calystegin B2
4	1.084	148.0976	–	C_6_H_13_NO_3_	Fagomine
7	1.562	130.0862	–	C_6_H_11_NO_2_	Pipecolic acid
7	1.243	–	191.0566	C_7_H_12_O_6_	Quinic acid
9	8.332	181.0725	–	C_7_H_8_N_4_O_2_	Theobromine
11	14.262	–	449.1110	C_21_H_22_O_11_	Astilbin
12	14.933	–	447.0950	C_21_H_20_O_11_	Quercitrin
14	15.752	–	431.0997	C_21_H_20_O_10_	Afzelin
**Ext-FL**					
4	1.030	176.0918	–	C_7_H_13_NO_4_	Calystegin B2
4	1.218	–	191.0567	C_7_H_12_O_6_	Quinic acid
7	1.261	138.0546	–	C_7_H_7_NO_2_	Trigonelin
8	11.490	–	289.0722	C_15_H_14_O_6_	Mikanolide
13	8.138	181.0724	–	C_7_H_8_N_4_O_2_	Theobromine
13	14.823	–	447.0945	C_21_H_20_O_11_	Quercitrin
15	10.535	195.0880	–	C_8_H_10_N_4_O_2_	Caffeine
17	11.786	291.0867	–	C_15_H_14_O_6_	Luteoforol
19	12.098	865.2003	–	C_45_H_36_O_18_	Cinnamtannin D1
25	14.609	551.1049	–	C_24_H_22_O_15_	Quercetin 3-O-malonylglucoside
**Ext-LR**					
3	0.979	–	174.0768	C_7_H_13_NO_4_	Calystegin B2
6	0.953	176.0918	–	C_7_H_13_NO_4_	Calystegin B2
7	1.124	148.0973	–	C_6_H_13_NO_3_	Fagomine
9	9.510	–	225.0760	C_11_H_14_O_5_	Genipin
11	10.559	–	577.1339	–	Procyanidin
12	11.134	–	289.0705	C_15_H_14_O_6_	Mikanolide

**Table 2 pharmaceuticals-17-00869-t002:** Quantification of phenolic compound content and antioxidant potential of *Paullinia cupana* aqueous extracts.

Extract	TPC	DPPH	ABTS
Ext-LD	437.5 ± 0.093	40.57 ± 0.038	19.24 ± 0.003
Ext-FL	646.8 ± 0.165	40.37 ± 0.008	19.26 ± 0.002
Ext-LR	728.4 ± 0.087	41.72 ± 0.023	19.24 ± 0.002

Results expressed in µg GAE/g. Values are represented as mean ± standard deviation of triplicate experiments.

**Table 3 pharmaceuticals-17-00869-t003:** Colloidal stability of AgNPs for one day (D1), seven days (D7), and thirty days (D30) with values of hydrodynamic diameter (DH), polydispersity index (PdI), and surface zeta potential (ZP) after storage at room temperature (RT) or in a refrigerator (REF).

Time/Storage	AgNPs-LD		
	HD (nm)	PdI	ZP (mV)
D0	79.29 ± 17.40	0.303 ± 0.057	−28.7 ± 0.80
D1—RT	69.70 ± 9.10	0.262 ± 0.065	−30.8 ± 2.20
D1—REF	79.78 ± 15.40	0.285 ± 0.089	−35.2 ± 1.20
D7—RT	61.98 ± 8.04	0.358 ± 0.091	−18.4 ± 5.35 ^×^
D7—REF	66.61 ± 1.67	0.406 ± 0.022	−18.4 ± 2.26 ^×^
D30—RT	77.29 ± 2.64	0.441 ± 0.028	−39.1 ± 0.32 ^×^
D30—REF	74.28 ± 4.24	0.479 ± 0.062	−36.6 ± 1.14 ^×^
**Time/Storage**	**AgNPs-FL**		
	**HD (nm)**	**PdI**	**ZP (mV)**
D0	78.87 ± 4.10	0.311 ± 0.075	−33.8 ± 0.50
D1—RT	70.74 ± 2.70	0.324 ± 0.069	−26.6 ± 1.90 *
D1—REF	73.85 ± 4.40	0.310 ± 0.078	−36.0 ± 2.40
D7—RT	63.66 ± 1.46 *	0.280 ± 0.012	−39.7 ± 3.72
D7—REF	68.54 ± 1.74 *	0.335 ± 0.039	−37.9 ± 1.40
D30—RT	66.60 ± 0.97 *	0.277 ± 0.008	− 39.4 ± 1.42
D30—REF	70.25 ± 5.03	0.392 ± 0.091	−38.2 ± 2.73
**Time/Storage**	**AgNPs-LR**		
	**HD (nm)**	**PdI**	**ZP (mV)**
D0	86.76 ± 9.90	0.417 ± 0.036	−38.5 ± 1.40
D1—RT	78.63 ± 6.00	0.347 ± 0.028	−34.5 ± 0.40
D1—REF	85.20 ± 10.20	0.311 ± 0.032 ^ϕ^	−35.2 ± 0.30
D7—RT	71.69 ± 2.89	0.322 ± 0.041 ^ϕ^	−36.7 ± 3.11
D7—REF	77.55 ± 0.99	0.374 ± 0.017	−37.8 ± 0.89
D30—RT	101.6 ± 19.07	0.299 ± 0.044 ^ϕ^	−36.0 ± 1.06
D30—REF	81.52 ± 1.99	0.350 ± 0.025	−16.8 ± 2.34 ^ϕ^

Values are represented as mean ± standard deviation of the mean of measurements in triplicate. Statistical analysis: One-way ANOVA test (*p* < 0.05), followed by Tukey’s test. Superscript symbols indicate significant differences within each parameter separately for each AgNPs group compared to day 0 (D0) measurement. ^×^ represents the statistical differences for AgNPs-LD, * represents the statistical differences for AgNPs-FL, and ^ϕ^ represents the statistical differences for AgNPs-LR.

**Table 4 pharmaceuticals-17-00869-t004:** Hydrodynamic diameter and concentration of AgNPs synthesized from extracts of *Paullinia cupana* plant parts obtained by nanoparticle tracking analysis.

Samples	Diameter (nm)	Concentration (Particles/mL)
AgNPs-LD	68.9 ± 0.7	1.56 × 10^8^
AgNPs-FL	61.4 ± 1.0	1.68 × 10^11^
AgNPs-LR	78.4 ± 2.6	1.43 × 10^10^

The values are represented as the mean ± standard deviation of the mean of triplicate measurements.

**Table 5 pharmaceuticals-17-00869-t005:** Antimicrobial activity of AgNPs obtained from aqueous extracts of *Paullinia cupana* leaves and flowers against bacteria and fungi.

Samples	Microorganisms					
	*A. fumigatus*		*C. albicans*		*E. coli*	
	MIC	MBC	MIC	MBC	MIC	MBC
**AgNPs-LD**	2.12	–	2.12	–	2.65	2.65
**AgNPs-FL**	2.12	–	4.25	–	5.3	5.3
**AgNPs-LR**	2.6	–	5.3	–	10.6	10.6
**AgNO_3_**	1.3	–	1.3	–	10.6	21.25
	** *F. oxysporum* **		** *P. chrysogenum* **		** *S. aureus* **	
	MIC	MBC	MIC	MBC	MIC	MBC
**AgNPs-LD**	4.25	–	2.12	–	10.6	21.25
**AgNPs-FL**	4.25	–	2.12	–	10.6	21.25
**AgNPs-LR**	21.25	–	2.6	–	10.6	10.6
**AgNO_3_**	1.3	–	2.6	–	10.6	10.6

MIC and MBC: µg/mL.

## Data Availability

The data presented in this study are available on request from the authors.

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
