# Peer review of "Synthesis of Silver Nanoparticles Using Extracts from Different Parts of the Paullinia cupana Kunth Plant: Characterization and In Vitro Antimicrobial Activity"

_pharmaceuticals, 2024, doi:10.3390/ph17070869_

Round 1
Reviewer 1 Report
Comments and Suggestions for Authors
In this work, the authors proposed new methods for green synthesis of silver nanoparticles. The study represents an extensive study of the formation of gold nanoparticles using three P. Cupana extracts as reducing agents. The manuscript is consistent and easy to read. It's positive that the study includes the evaluation of influence of variable parameters (seasonality on the collection of biological material, using different parts of the same plant) on the synthesized nanoparticles and also contains the comparison of the nanoparticles properties obtained with described technique with silver NPs prepared using the leaves from the other plants. In general, the work evokes a good impression and only small issues should be improved.
I have only two questions and two notes for the authors.
1. What can explain the decrease in the diameter of AgNPs-FL after one week and one month storage compared to the fresh dispersions?
2. Why silver nitrate was used as a control during assessment of antimicrobial activity? Use of silver nanoparticles obtained with conventional chemical routes would be more honest.
3. Text in all figures should be larger, especially the scale mark on the TEM images.
4. One abbreviation (CFT or TPC) should be left for total phenol content.
Author Response
Comment 1. What can explain the decrease in the diameter of AgNPs-FL after one week and one month storage compared to the fresh dispersions?
Answer 1: Dear reviewer, thank you very much for your question. Our results indicate that the reducing potential of the Paullinia cupana flower extract contributed to a reduction in the diameter of the particles, as well as having a positive influence on the increase, in modulus, of the Zeta potential of the particles with the passing of the storage time, which is an indication of greater stability. Moreover, surface alterations, such as sorption/desorption of molecules and even corrosion of the silver core could contribute to this phenomenom. In addition, some studies have reported that the concentration of secondary metabolites in plant reproductive organs is higher than in vegetative organs and that these biomolecules can combine to provide a highly stable structure on the surface of AgNPs. The studies that can be consulted for these statements can be found at the links below:
Phytochemical screening of the leaf and flower Extracts of five ipomoea species collected From in and around Bangalore. DOI: 10.22376/2016.7.4.p71-73
Plant secondary metabolites in nectar: impacts on pollinators and ecological functions https://doi.org/10.1111/1365-2435.12761
Secondary metabolites in plants: transport and self-tolerance mechanisms https://doi.org/10.1080/09168451.2016.1151344
Comment 2. Why silver nitrate was used as a control during assessment of antimicrobial activity? Use of silver nanoparticles obtained with conventional chemical routes would be more honest.
Dear reviewer, we understand your question, but we tested silver nitrate as a positive control because it is the conventional control in the literature, as the antimicrobial activity of silver nanoparticles is usually accepted to be a consequence of the release of Ag+ cations. Moreover, there are many different methods and formulations of silver nanoparticles, a fact that would make it difficult to choose one single AgNP control. By comparing our formulations with the aqueous solution of this metal cation, we can, in a way, understand the antimicrobial effect of our particles on a nanometric scale and thereby increase the potential of AgNPs synthesized by safe routes using Paullinia cupana extracts.
3. Text in all figures should be larger, especially the scale mark on the TEM images.
Dear reviewer, thanks for the suggestion. We have made the changes to the texts of the UV/Vis and TEM figures (figures 1 and 2, respectively).
4. One abbreviation (CFT or TPC) should be left for total phenol content.
Dear reviewer, thank you very much for your correction. We have corrected this abbreviation by highlighting it in yellow in the text of the manuscript.

Reviewer 2 Report
Comments and Suggestions for Authors
Please, see document attached

Manuscript needs to be English check and corrected to improve several parts, but specially the abstract and introduction. Also, I’m not sure that the tittle is grammatically correct.
Author Response
Comment 1. Manuscript needs to be English check and corrected to improve several parts, but specially the abstract and introduction. Also, I’m not sure that the tittle is grammatically correct. Amazon is not an appropriate keyword
Answer 1. Dear reviewer, we have revised the language in our manuscript as advised. The title was changed. The keyword Amazon was replaced by "green synthesis".
Comment 2. There are some typos and formatting issues. For instance: AgNO3 line602, FIGURE instead of Figure line 210, and some reference in yellow highlight.
Answer 2. Dear reviewer, thank you very much for your comment regarding the errors and words in the lines mentioned. We have analyzed your suggestions and corrected what we could by highlighting them in yellow in the text of the manuscript.
Comment 3. The number of citations looks excessive for a research article. Author should cite
the most significant and up to date references.
Dear reviewer, thank you for your comment. However, we believe that, as this is a research article, our choice to discuss the results obtained is conducted based on old and current citations to highlight our main scientific observations, which is why the number of references cited is justifiable.
Comment 4. In the introduction, line 105 authors indicate that it is the first time that guarana leave and flower extracts are used for the synthesis of silver nanoparticles. However, the same authors have published this year the article: Phytosynthesis of silver nanoparticles using guarana (Paullinia cupana Kunth) leaf extract employing different routes: characterization and investigation of in vitro bioactivities.
Answer 4. Dear reviewer, in fact we recently published this article and used Paullinia cupana leaves as a source of plant metabolites for the biosynthesis of AgNPs. We are sorry for our mistake, and we thank you for your careful analysis. We have removed the expression "for the first time" from line 105. Thank you very much.
Comment 5. Authors needs to compare how both syntheses are different and compare the results between them. Not just the citation that appears regarding the composition characterization of the extract.
Answer 5. Dear reviewer, thank you very much for your suggestion. In writing our manuscript, we evaluated the comparisons made in relation to the syntheses and we believe that the results of each of them are well discussed in relation to both the characterization and the application of biogenic AgNPs and plant extracts.
Comment 6. Why the extraction is so short? Just 3 min at boiling temperatures. Did you optimise the extractions conditions?
Answer 6. Dear reviewer, in a brief explanation, this manuscript is part of the doctoral thesis of the main author, Alan Kelbis Oliveira Lima, and yes, in his complete study, the author optimized the extraction conditions, assessing that this boiling time was ideal for obtaining biomolecules capable of reducing metal ions, without having considerable losses of thermosensitive metabolites. In addition, after preparing the extract by boiling, the author also uses filtration to improve the characteristics of the aqueous extract, keeping only the biomolecules that have been extracted, without the presence of traces of the plant preparation. Thank you very much for the opportunity to explain a little more about our methodology.
Comment 7. Authors indicated that they extract mostly alkaloids, flavonoids, phenolic acids, terpenoids, and procyanidins. However, they do not analyse other water-soluble components such as amino acids, proteins, or carbohydrates.
Answer 7. Dear reviewer, in fact, based on the techniques used to characterize the aqueous plant extract of Paullinia cupana, as well as the databases used to confirm the metabolites, we did not characterize components such as proteins, carbohydrates and amino acids that could certainly be involved in biogenic synthesis as agents for reducing/stabilizing nanometric particles, as reported in previous studies characterizing the extract of this plant. Examples:
Seed oil composition of Paullinia cupana var. sorbilis (Mart.) Ducke https://doi.org/10.1007/s11745-003-1126-5
Guarana powder polysaccharides: Characterisation and evaluation of the antioxidant activity of a pectic fraction https://doi.org/10.1016/j.foodchem.2012.03.088
Health and technological aspects of methylxanthines and polyphenols from guarana: A review
https://doi.org/10.1016/j.jff.2018.05.048
However, in our research, we believe that the description of various other metabolites and classes of biomolecules present in the extracts of Paullinia cupana leaves and flowers responds to this crucial stage of characterizing the biological matrix used in biosynthesis and can elucidate, even partially, probable mechanisms of synthesis and stability of AgNPs. Another highlight is the lack of information on the phytochemical composition of these plant parts of Paullinia cupana published in previous studies, so our results provide important support in maintaining this information for the scientific community.
Comment 8. Authors perform the synthesis of silver nanoparticles at 70ºC for 3 hours. However, in the discussion they indicated that the synthesis continues for 30 days since they observed an increase in the absorbance. Authors should optimise the synthesis conditions, so the reaction is complete at a certain time. How can you use this nanoparticle for biological activity if they vary with the time when you perform the analysis? Did you just perform one synthesis with one condition? Did you not optimise the reactions conditions to attain homogeneous size and shape?
Answer 8. Dear reviewer, thank you very much for your comments. They are pertinent and require a more comprehensive explanation since we believe it was an error to write that the nucleation process was still taking place, since it is not possible to know whether this phenomenon was in fact still happening. We have therefore removed the expression "indicating that the nucleation process was still tak-ing place" from lines 219 and 220 of the original document.
As previously mentioned, the complete study is part of Alan Kelbis Oliveira Lima's doctoral thesis and previous optimizations were made both in the way the extract was prepared, with a view to better extraction, and in the parameters involved in the synthesis of AgNPs. In our current study, we chose to publish our results from both the synthesis using Paullinia cupana leaves and flowers and we observed, according to the results in Table 3, that the colloidal characteristics of the AgNPs after 30 days remain, in general, like the characteristics obtained in the initial syntheses. In addition, according to figure 2, the morphology and dry diameter also follow similar patterns between the different groups of AgNPs.
Comment 9. Please, correct Figure 1A, part of the graph is missing.
Answer 9. Dear proofreader, we believe that the original file was badly formatted. We are sorry for that. We have corrected figure 1A by increasing the size of the text. Thank you.
Comment 10. In table 3, the hydrodynamic size presents a high standard deviation which could indicate that the measurements were not performed with the best conditions or that some of the replicates needs to be excluded.
Answer 10. Dear reviewer, we agree with your comment about the high deviations observed. We are sure that our measurements were carried out under the same conditions, care and standards that surround our work in the laboratory, but as this is an overly sensitive analysis technique from the point of view of results, we also agree that these variations can sometimes occur, which does not belittle or take away from the merit of our study, but which certainly makes us more attentive to such analyses in future publications.
Comment 11. On the other hand, in the TEM images authors present standard deviations on the mean size really small that do not much with the nanoparticles observed in the images nor with the size distribution observed in the histogram.
Answer 11. Dear reviewer, your observation is indeed remarkably interesting from the point of view of carefully analyzing our study. Our research was based on the rule of the same conditions of analysis in particle counting and these analyses were carried out by the same researcher, on the same day, which eliminates probable biases of our results.
About the TEM images, we chose to include in the manuscript those that clearly showed the morphology of the particles, as well as those that showed different morphologies in the same sample area. In addition, we would like to point out that the size distribution histogram reflects the particle count of a larger group of samples than that shown in images 2A, 2B and 2C, which may give the impression of possible errors in the deviation, size, and distribution of the AgNPs. We hope we have satisfactorily explained your doubts.
Comment 12. In Table 4 they present size and concentration obtained for each sample. It can be noted that the first one presents the smallest size and the smallest concentration while the third one presents the biggest size and the higher concentration. If the synthesis was performed in the same conditions in the three cases, this implies that there are Ag(I) which not reacted. Did you perform some purification before the antimicrobial assays to eliminate the unreacted Ag? Authors report 3 different techniques to study the size of the nanoparticles, however they did not characterize other aspects such as crystallinity.
Answer 12. Dear reviewer, thank you for your comments on the aspects involved in our characterization of AgNPs. Regarding our table 4, the first sample - AgNPsLD - does have the lowest concentration of particles, but not the smallest diameter, while the third sample - AgNPs-LR - has the largest diameter, but not the smallest concentration. We are confirming this data because this is how it is shown in table 4 of our original document and to avoid errors if the document sent to you is different. In our current study, we did not purify the AgNPs before the antimicrobial assays to eliminate the unreacted Ag+ ions, but we will address this aspect and look for research partnerships that master the XRD technique in our subsequent studies with the aim of improving the purity and applications of our nanostructures and respecting your comment and observation.
Comment 13. Also, I encourage the characterization of the extracts by FTIR before and after the synthesis and the UHPLC-HRMS/MS after the synthesis to obtain more information regarding the possible synthetic mechanisms.
Answer 13. Dear reviewer, thank you very much for your tips. We strongly guarantee that our future studies perform with FTIR and UHPLC-HRMS/MS analyses of the plant extracts after synthesis to better elucidate the mechanisms that are still so unclear regarding the biosynthesis of metallic nanostructures, including AgNPs. We admit that we did not have such a thought during the development of this research.
Comment 14. Regarding the discussion of the antimicrobial assays, the samples are only more effective in E.coli than the AgNO3. In the other microorganisms, the effect of AgNO3 is higher. With just the MIC values authors made an extensive discussion on mechanism of action of AgNPs that are not corroborated in this study.
Answer 14. Dear reviewer, we very much appreciate your comments and the opportunity to comment further on the topic of antimicrobial activity. This application was explored in our study to corroborate many other studies published in the journal Pharmaceuticals in which the antimicrobial action of biogenic AgNPs was highlighted using various plant extracts, as shown in the examples below:
Exploring the Antimicrobial, Antioxidant, and Antiviral Potential of EcoFriendly Synthesized Silver Nanoparticles Using Leaf Aqueous Extract of Portulaca oleracea L. https://doi.org/10.3390/ph17030317
Euphorbia royleana Boiss Derived Silver Nanoparticles and Their Applications as a Nanotherapeutic Agent to Control Microbial and Oxidative Stress-Originated Diseases. https://doi.org/10.3390/ph16101413
Size-Dependent Antibacterial, Antidiabetic, and Toxicity of Silver Nanoparticles Synthesized Using Solvent Extraction of Rosa indica L. Petals https://doi.org/10.3390/ph15060689
In addition, to correlate the biological effect of nanostructures synthesized using Amazonian plants, increasing the potential for using Brazilian flora, we explored different parts of Paullinia cupana and made the study quite innovative. Therefore, given that our results varied depending on the microorganism tested and the type of sample, we have chosen to discuss the general mechanisms by which AgNPs can cause toxicity to microbial cells, always indicating citations that support our discussion and that have described these mechanisms for the different types/species of organisms tested and in view of the number of studies currently published that are dedicated to explaining these effects.
In addition, we correlated the physicochemical and morphological characteristics of AgNPs-LD, AgNPs-FL and AgNPs-LR with the probable antimicrobial mechanisms mentioned, even suggesting that our nanostructures have potential applications in various sectors of industry. We reiterate that the aim was to discuss the effects obtained, without judging the best or worst results, even though this information has been described in our discussion and in the conclusions section of the manuscript.
Comment 15. In the conclusions, authors indicated that “In addition, the AgNPs exhibited significant antimicrobial action against various bacterial and fungal strains, surpassing that of the positive control (aqueous AgNO3 solution) and regardless of the time the plant material collection and the part of the plant.” This do not match the results observed in Table 5.
Answer 15. Dear reviewer, thank you for your comment and observation. In fact, in the passage you mentioned, we implied that the different groups of AgNPs had a greater antimicrobial effect than the positive control (AgNO3), which is not consistent with what is described in Table 5, where only the bacteria E. coli and the fungus P. chrysogenum had this effect improved, and for the bacteria S. aureus this effect was the same as the positive control. We have therefore changed the wording of the expression mentioned in the conclusion to give the most correct information possible about the effect observed.
Reviewer 3 Report
Comments and Suggestions for Authors
The manuscript reports the green synthesis of Ag NPs using aqueous extracts of guarana (Paullinia cupana) leaves and flowers and evaluates their in-vitro antimicrobial activity. Green synthesis of NPs using plant extract has received increased attention in the last few decades due to their outstanding biomedical applications and environmentally friendly nature. The present work affords a promising strategy for producing green Ag NPs for antibacterial applications. The manuscript contains grammatical mistakes and has certain limitations. Therefore, I suggest major revisions for quality enhancement, which are given below.
1. The manuscript contains many grammatical and typing mistakes even in the title, which must be carefully read and removed, and the language needs sufficient improvement.
2. Discuss some biological applications of Ag NPs by using and citing these latest articles; https://doi.org/10.2147/ijn.s453775
3. Mention the toxicity and effects of the selected bacterial strains in the introduction section.
4. The quality of Figure 1 needs improvement.
5. Carefully read all the tables and values for any errors.
6. Authors are suggested to perform XRD analysis for evaluating the crystalline nature of the particles.
7. Authors are suggested to perform FTIR analysis to analyze the various functional groups present on the green synthesized NPs.
8. Represents antibacterial photographs in the manuscript as support and proof of the antibacterial activity of the Ag NPs.
9. Draw a mechanism diagram for the explanation of the antibacterial explanation.
10. There is no need to explain the details of the details of TEM and EDX analysis in section 3. Minimize it.
11. Insert as materials section in section 3 and include all the chemicals used and their specifications.
12. Discuss the obtained scientific results in the conclusion section.
Comments on the Quality of English LanguageExtensive editing of English language required.
Author Response
Comment 1. The manuscript contains many grammatical and typing mistakes even in the title, which must be carefully read and removed, and the language needs sufficient improvement.
Answer 1. Dear reviewer, thank you very much for pointing out the grammatical and typing errors. We have once again curated the language and likely words/expressions and hope that the new writing is improved. Some yellow highlighting has been done in the text to exemplify some of the specific changes that have been made.
Comment 2. Discuss some biological applications of Ag NPs by using and citing these latest articles; https://doi.org/10.2147/ijn.s453775
Answer 2. Dear reviewer, the study you suggested was especially important for our list of references. We have added this research to our manuscript. Thank you very much!
Comment 3. Mention the toxicity and effects of the selected bacterial strains in the introduction section.
Answer 3. Dear reviewer, thank you for your suggestion and comment. These mentions of microorganisms are described in the discussion of antimicrobial activity. Thank you very much!
Comment 4. The quality of Figure 1 needs improvement.
Answer 4. Dear reviewer, we have looked at the quality of our figures and realized that the formatting may have been changed and that is why the quality of some of them is lower than the form we sent to the journal. We have improved the quality of the figures, including increasing the text of the caption. Thank you very much for your suggestions.
Comment 5. Carefully read all the tables and values for any errors.
Answer 5. Dear reviewer, we have once again analyzed our tables and values in detail. We guarantee that the figures are in line with our results. Thank you very much for the opportunity to reaffirm our data.
Comment 6. Authors are suggested to perform XRD analysis for evaluating the crystalline nature of the particles.
Answer 6. Dear reviewer, thank you very much for suggesting the use of the XRD technique. Now we are unable to carry it out and include it in this manuscript, but we believe it is important to evaluate the crystal structure of AgNPs in our work and we strongly hope to have such results in the future.
Comment 7. Authors are suggested to perform FTIR analysis to analyze the various functional groups present on the green synthesized NPs.
Answer 7. Dear reviewer, we would also like to thank you for your suggestion to include FTIR analysis of our biogenic AgNPs. Now, for our current manuscript, such an analysis is unfeasible from the point of view of the time it takes to send the revised manuscript.
We are sure that evaluating the functional groups present in nanostructures is important from a characterization point of view, which is why the description of the UHPLC-HRMS/MS results is included in our document for publication in your journal.
Comment 8. Represents antibacterial photographs in the manuscript as support and proof of the antibacterial activity of the Ag NPs.
Answer 8. Dear reviewer, thank you for your comment. Our researchers specializing in antimicrobial tests carried out the in vitro tests using the microdilution method, visually observing the presence/absence of turbidity in the culture medium, which reflected the antimicrobial action of our treatments. Therefore, no photographs were taken of the results. We very much hope that you understand this, and that we can ensure that the experiments are carried out with the necessary care that publication in your journal requires.
Comment 9. Draw a mechanism diagram for the explanation of the antibacterial explanation.
Answer 9. Dear reviewer, thank you for your comment. We have evaluated your suggestion and decided to retain the text explanation of the probable mechanisms of antimicrobial activity. These mechanisms are general ways in which AgNPs can affect bacterial and fungal cells, but we did not specifically investigate these mechanisms in our study. In addition, as we worked with different microorganisms in our study, we believe that we need to specialize in studying these mechanisms for different strains, and further studies are necessary for this. Thank you very much for the opportunity to explain our justifications and in future studies the graphic design of the mechanisms can be added.
Comment 10. There is no need to explain the details of the details of TEM and EDX analysis in section 3. Minimize it.
Answer 10. Dear reviewer, thank you for your comment and suggestion. We think that in section 3 the details of the TEM and EDX analyses are sufficient and allow for future reproduction of your methods by other researchers.
Comment 11. Insert as materials section in section 3 and include all the chemicals used and their specifications.
Answer 11. Dear reviewer, thank you for your suggestion. We have added this section to the "material and methods" topic in our research document.
Comment 12. Discuss the obtained scientific results in the conclusion section.
Answer 12. Dear reviewer, thank you very much for your comment. We have made changes to the conclusion section in our manuscript and have highlighted it in yellow.
Round 2
Reviewer 2 Report
Comments and Suggestions for Authors
Authors gave a rational answer to all my previous concerns.
Author Response

(The authors gave the same response as above.)

Reviewer 3 Report
Comments and Suggestions for Authors
Manuscript is sufficiently improved however most of the comments are not addressed.
1. The number of figures are very less. Include FTIR or mechanism of antibacterial activity.
2. The supplementary materials is authorship contribution form. What is this?
Comments on the Quality of English LanguageThe language needs minor checking
Author Response

(The authors gave the same response as above.)
